# Preventive Effect of Probiotics on Oral Mucositis Induced by Cancer Treatment: A Systematic Review and Meta-Analysis

**DOI:** 10.3390/ijms232113268

**Published:** 2022-10-31

**Authors:** Yu-Cheng Liu, Chia-Rong Wu, Tsai-Wei Huang

**Affiliations:** 1School of Medicine, College of Medicine, Taipei Medical University, 250 Wu-Hsing Street, Taipei 110, Taiwan; 2School of Nursing, College of Nursing, Taipei Medical University, Taipei 110, Taiwan; 3Center for Nursing and Healthcare Research in Clinical Practice Application, Wan Fang Hospital, Taipei Medical University, Taipei 116, Taiwan; 4Cochrane Taiwan, Taipei Medical University, Taipei 110, Taiwan; 5Department of Nursing, Wan Fang Hospital, Taipei Medical University, Taipei 116, Taiwan

**Keywords:** probiotics, lactobacillus, oral mucositis, cancer, chemotherapy, radiotherapy

## Abstract

Oral mucositis is a common adverse effect of cancer therapy. Probiotics have been shown to exert anti-inflammatory and immunomodulatory effects. We performed a meta-analysis of randomized controlled trials (RCTs) to investigate whether probiotics can prevent cancer therapy–induced oral mucositis. We searched PubMed, Embase, Cochrane Library, and ClinicalTrials.gov databases for trials related to probiotics and oral mucositis published before September 2022; no language restrictions were applied. The primary outcome was the incidence of oral mucositis and severe oral mucositis. Secondary outcomes were the requirement for enteral nutrition during treatment, body weight loss, and decreased quality of life. The study has been registered in PROSPERO (number: CRD 42022302339). Eight RCTs, including 708 patients, were reviewed; however, a meta-analysis of only seven trials could be performed. Three trials using Lactobacilli-based probiotics reported that the incidence of oral mucositis in the probiotic group was significantly low (risk ratio [RR] = 0.84, 95% confidence interval [CI] = 0.77–0.93, *p* = 0.0004). Seven trials reported a significantly low incidence of severe oral mucositis in the probiotic group (RR = 0.65, 95% CI = 0.53–0.81, *p* < 0.0001). The requirement of enteral nutrition was significantly low in the probiotic group (odds ratio = 0.34, 95% CI: 0.13–0.92, *p* < 0.05). This study demonstrated the effectiveness of probiotics in the prevention and mitigation of cancer therapy–induced oral mucositis. We recommend the use of probiotics to prevent and treat oral mucositis during cancer therapy.

## 1. Introduction

Oral mucositis is a common adverse effect of cancer treatment, including chemotherapy, radiotherapy, concurrent chemoradiotherapy, and hematopoietic stem cell transplantation (HSCT) [1]. Oral mucositis, characterized by erythematous and ulcerative lesions of the oral mucosa, is caused by the production of reactive oxygen species and subsequent cellular damage [2]. The incidence of oral mucositis is approximately 20–40% in patients undergoing chemotherapy, 60–85% in patients receiving HSCT, and up to 90% in patients with head and neck cancer undergoing chemoradiotherapy [3]. Oral mucositis is a painful condition that can cause poor oral hygiene, difficulty eating and swallowing, altered nutritional intake, and decreased quality of life [4,5]. Severe oral mucositis can lead to a reduction or discontinuation of cancer treatment, resulting in a poor prognosis [5,6]. Therefore, effective strategies for the prevention of oral mucositis are necessary.

According to the guidelines of the Multinational Association of Supportive Care in Cancer (MASCC)/International Society of Oral Oncology mucositis, various treatment options, such as photobiomodulation therapy and natural and pharmacological agents, are beneficial for the prevention of oral mucositis secondary to cancer treatment [1]. Compared to synthetic drugs, natural remedies are typically considered to have fewer side effects and are more accessible for patients in resource-restricted geographical regions [7]. Honey, which can be applied topically and administered systemically, is the only natural agent suggested for the prevention of oral mucositis in patients with head and neck cancer who receive radiation therapy or chemoradiotherapy [7]. However, the daily use of honey for a long period is concerning because of its cariogenic effect and its potential to cause dental caries [8,9,10]. The use of probiotics, another natural agent with possible benefits in the prevention of oral mucositis after cancer therapy, has not been suggested by MASCC due to limited evidence [7].

Probiotics are living microorganisms that provide health benefits to the host when administered in sufficient numbers [11], lactobacilli and bifidobacterium being the two most widely used bacterial genera as probiotics [12]. Probiotics have been shown to exert anti-inflammatory and immunomodulatory effects [13,14]. In recent years, the use of probiotics in cancer care-related fields have gained increasing attention because of their possible benefits in the prevention of cancer therapy-induced toxic side effects, including diarrhea and oral mucositis [15]. A large double-blind, randomized controlled trial reported that lactobacillus spp. probiotics significantly reduced the incidence and severity of diarrhea in patients with sigmoid, rectal, and cervical cancer who received pelvic radiotherapy [16]. Furthermore, another study demonstrated the protective effect of probiotics against oral mucositis in patients with nasopharyngeal carcinoma receiving chemoradiotherapy [17]. The use of probiotics is generally considered safe [18,19], in addition to being beneficial

Recently, Feng et al. performed a meta-analysis to investigate the efficacy of probiotics in cancer therapy-induced oral mucositis [20]. According to their findings, probiotics exerted a significant protective effect against oral mucositis in cancer patients. However, there was still a lack of evidence supporting the use of certain bacterial species or strains as probiotics against cancer therapy-induced oral mucositis, and no previous studies had provided conclusive evidence of the effectiveness of probiotics during certain cancer treatments with a high incidence of oral mucositis, such as chemoradiotherapy. Since new evidence may emerge, we conducted an updated systematic review and meta-analysis of randomized controlled trials (RCTs) to further evaluate the preventive and therapeutic effects of probiotics in the development of oral mucositis induced by cancer therapy.

## 2. Results

### 2.1. Characteristics of Trials

Figure 1 presents the flowchart of the screening and selection of trials. Our initial search of four databases (PubMed, Embase, Cochrane Library, and ClinicalTrials.gov) and one additional source of the “related articles” option in PubMed yielded 385 trials. After removing duplicates, 348 trials were screened using titles and abstracts, with 310 trials excluded. Subsequently, after the full text evaluation of the remaining 38 trials, we excluded 30 trials for the following reasons: 11 were clinical trial registration records that did not provide results, four did not have their full text available, three were review articles, one was a case report, four were single-arm studies, one was a comment article, three were animal studies, two were articles focusing on different topics, and one was a trial applying different comparison (synbiotics vs. probiotics). Finally, we included eight trials in this meta-analysis [17,21,22,23,24,25,26,27]. All included studies were published in English.

Table 1 lists the baseline characteristics of these trials. The eight trials were published between 2007 and 2022 and included a total of 708 patients, with sample sizes ranging from 25 to 200. Six trials included patients with head and neck cancers [17,21,22,23,25,27], of which two trials only included patients with locally advanced nasopharyngeal carcinoma [17,27]; five of these trials [17,21,23,25,27] were administered chemoradiotherapy, while the remaining trial were administered only chemotherapy [22]. Topuz et al. [26] and Österlund et al. [24] included patients diagnosed with colorectal cancer, who received chemotherapy. In three trials, a combination of probiotics containing lactobacillus spp. and *Bifidobacterium* spp. was used [17,26,27]. Four trials used probiotics containing lactobacillus spp. only [21,22,24,25]. Mirza et al. [23] administered probiotics containing only *Bacillus clausii*. All included trials investigated whether the use of probiotics can prevent or reduce the severity of oral mucositis after cancer therapy. Oral mucositis was evaluated according to the National Cancer Institute (NCI) Common Terminology Criteria for Adverse Events (CTCAE) [28,29] or using the World Health Organization mucositis classification scale [30]. Severe oral mucositis was defined as grade 3 or higher oral mucositis.

Table 2 presents the methodological quality of the included trials. All of the trials included had a low risk of bias caused by the selection of reported results. Regarding bias caused by the inadequacy of randomization, four trials had a low risk [17,23,25,27], two trials exhibited some concerns due to lack of information on randomization [22,26], and the remaining two trials demonstrated a high risk due to their open-label design [21,24]. Seven trials had a low risk of bias in deviations from intended interventions [17,22,23,24,25,26,27], while one trial exhibited a high risk due to the high percentage of interruptions in interventions (34.3%) [21]. Seven trials had a low risk of bias caused by missing data from dropouts and outcome measurement [17,21,23,24,25,26,27], while the remaining trial had a high risk and some concerns due to the high percentage of missing data (24%) and the lack of information on blinding of the outcome assessor [22].

### 2.2. Primary Outcome

Seven trials [17,21,23,24,25,26,27] assessed oral mucositis according to the NCI CTCAE, and one trial [22] evaluated oral mucositis using the WHO mucositis grading scale. Four trials reported the incidence of oral mucositis [17,23,25,27], and seven trials [17,21,22,23,24,25,27] provided data on severe oral mucositis. The data provided by Topuz et al. [26] could not be pooled due to the different presentation of the results. The overall incidence of oral mucositis was 81.5% in the probiotic group and 96.3% in the control group; the incidence of severe oral mucositis was 34.3% and 56.6% in the probiotic group and the control group, respectively.

#### 2.2.1. Incidence of Oral Mucositis

Four of the eight trials provided the incidence of oral mucositis after cancer therapy (Figure 2). The results of the meta-analysis indicated that the probiotic group using a lactobacilli-based regimen demonstrated an insignificantly lower incidence of oral mucositis (RR = 0.84, 95% CI = 0.77–0.93, *p* = 0.0004), with low heterogeneity (I^2^ = 40%) in the trials analyzed.

#### 2.2.2. Incidence of Severe Oral Mucositis

Seven trials reported the incidence of severe oral mucositis (Figure 3a). The probiotic group showed a significantly lower incidence of severe oral mucositis (RR = 0.65, 95% CI = 0.53–0.81, *p* < 0.0001), with low heterogeneity (I^2^ = 16%) in all trials analyzed.

In the subgroup analysis, we investigated the effects of probiotics on patients receiving chemotherapy and determined whether the use of probiotics containing lactobacillus spp. alone can reduce the incidence of severe oral mucositis after cancer therapy.

Five trials in our meta-analysis included patients receiving chemo-radiotherapy, whereas the remaining trials included patients receiving chemotherapy only; the use of probiotics significantly reduced the risk of severe oral mucositis, with a RR of 0.61 (95% CI = 0.46–0.82, *p* = 0.001) and low heterogeneity (I^2^ = 40%; Figure 3b).

Four trials in our meta-analysis administered probiotics containing lactobacillus spp. only, while the others used a combination of probiotics or other bacterial species. The incidence of severe oral mucositis was significantly low (RR = 0.73, 95% CI = 0.60–0.88, *p* = 0.0009) in the probiotic group after the data were pooled, with low heterogeneity (I^2^ = 0%) observed in all four trials (Figure 3c).

Topuz et al. [26] reported the incidence of oral mucositis based on each chemotherapy course, but not on each patient. The incidence of oral mucositis did not differ significantly between the probiotic and control groups (27.3% vs. 21.7%, *p* > 0.05) during chemotherapy courses.

### 2.3. Secondary Outcomes

#### 2.3.1. Requirement of Enteral Nutrition

One trial compared the requirement for enteral nutrition between the probiotic and control groups during cancer treatment [21]. Compared to the control group, the probiotic group demonstrated a significant reduction in enteral nutrition requirement (OR = 0.34, 95% CI = 0.13–0.92, *p* < 0.05).

#### 2.3.2. Body Weight Loss

Three trials documented the body weight status of patients weekly during cancer treatment [17,21,27]. However, accurate data were not provided in all three trials, resulting in the infeasibility of quantitative analysis; we performed qualitative analysis. Both Jiang et al. [17] and Xia et al. [27] reported no significant differences in body weight reduction between the two groups (6.11% vs. 6.11%, *p* > 0.05 and 6.53% vs. 6.7%, *p* > 0.05, respectively). Sanctis et al. [21] reported that both groups exhibited a significant tendency (*p* < 0.01) to lose body weight compared to the loss at baseline during treatment; however, the comparison between the two groups was not described in the original article.

#### 2.3.3. Quality of Life

Two trials recorded patients’ QoL in two arms before and after cancer treatment [21,25], and both studies examined QoL by using the Functional Assessment of Cancer Therapy-Head and Neck (version 4.0) QoL questionnaire [31]. The results could not be used in quantitative analysis because of the lack of accurate QoL scores in both the original articles; hence, we instead performed qualitative analysis. Both trials did not report a significant effect on improving quality of life in the probiotic group compared to the control group.

## 3. Discussion

The results of the primary outcomes in our study demonstrated that probiotics significantly reduced the incidence of oral mucositis and severe oral mucositis following cancer treatment, which were consistent with previous studies [20,32]. In subgroup analysis, our findings revealed that probiotics significantly reduced the incidence of severe oral mucositis in patients receiving chemoradiotherapy, and the use of probiotics containing single strain lactobacilli was adequate to reduce the incidence of severe oral mucositis following cancer treatment. Regarding secondary outcomes, probiotics significantly reduced enteral nutrition requirement during treatment in patients with cancer; however, compared to the control group, probiotics did not show significant improvements in body weight loss and QoL during cancer treatment course.

Oral mucositis is a common adverse effect of cancer treatment, including radiotherapy and concurrent chemoradiotherapy. Direct damage to the oral mucosa caused by these cancer treatments leads to DNA damage, accumulation of reactive oxygen species, and epithelial cell death [33]. These factors lead to the activation of innate immune responses through various pathways, such as the nuclear factor kappa-B (NF-κB) pathway [34], leading to the production of pro-inflammatory cytokines including interleukin-6, tumor necrosis factor-alpha and oxidative stress responders, as well as local tissue inflammation [35]. With the progression of inflammation, ulceration may occur, leading to more immune responses and inflammatory processes, worsening mucosal damage [33]. However, the microbiota may play a role in the development of oral mucositis, as microbial dysbiosis, invasion, and colonization of the oral mucosa are found to be involved in the pathophysiology of oral mucositis [36].

Definite mechanisms through which probiotics prevent or mitigate cancer treatment–induced oral mucositis remain unclear. Several animal studies have attempted to explain the mechanisms underlying the development of oral mucositis. In mice, lactobacillus reuteri exerted a protective effect on 5-fluorouracil-induced oral mucosal damage [37]; lactobacillus reuteri not only downregulated NF-κB activation and pro-inflammatory cytokine expression, but also mediated the antioxidative effect through nuclear factor erythroid 2-related factor 2, a key transcription factor in mitigating oxidative stress [38]. Another animal study reported that Streptococcus salivarius K12 exerted a preventive effect on radiotherapy-induced oral mucositis in mice by modulating oral microbiota and improving microbial dysbiosis, which is involved in the development and progression of oral mucositis following radiotherapy [39]. Furthermore, a study using a mouse model demonstrated that lactobacillus spp. exerted a radioprotective effect on intestinal mucosal damage through the toll-like receptor 2 pathway, which involves the activation of cyclooxygenase-2-expressing mesenchymal stem cells and the production of prostaglandin E2, thus protecting against the radiation-induced apoptosis of epithelial stem cells [40]. One of the trials [27] included in this study conducted animal experiments to elucidate mechanisms underlying the development of mucositis induced by chemotherapy and radiotherapy; the study reported that the combination of probiotics downregulated the inflammatory signaling pathway, reduced epithelial cell apoptosis, and restored the disturbed microbial diversity in the gut following chemotherapy and radiotherapy [27]. In summary, the use of probiotics can prevent or mitigate oral mucositis by reducing inflammatory or oxidative responses, modulating and improving the dysregulation of the oral microbiota, and promoting epithelial cell protection. However, the underlying mechanisms of various bacterial species or strains in the prevention and mitigation of oral mucositis may be different [37], and all the mentioned studies were based on animal models. Therefore, the possible mechanisms should be interpreted with caution, and additional studies focusing on humans are warranted.

The findings of our study revealed the effective use of lactobacilli-based probiotics in the prevention and mitigation of oral mucositis after cancer treatment. Lactobacilli are gram-positive, facultative anaerobic or microaerophilic, rod-shaped, and non-spore-forming bacteria [41]. These lactic acid bacteria can ferment hexose sugars to produce lactic acid, resulting in the development of an acidic environment and inhibition of the growth of other bacterial species [41]. The lactobacillus genus comprises more than 200 species and is the normal flora found in the gastrointestinal tract and vagina in humans [42]. Lactobacillus species are commonly encountered in daily life because they are widely used to produce foods such as yogurt, cheese, and other fermented products [43]. In addition to their use in food production, lactobacillus species have also been found to have multiple medical applications. In the healthy population, lactobacilli use could reduce abnormal vaginal discharge (lactobacillus crispatus), treat periodontitis (*L. reuteri*), and reduce negative thoughts associated with sad mood in adults [44,45,46]. In patients with cancer, lactobacilli may help reduce the side effects induced by cancer treatments, including diarrhea and oral mucositis, as shown in our study [15]. In the future, lactobacilli may play a crucial role in the treatment or even prevention of cancer because they appear to be capable of modulating immune responses and causing direct cytotoxic effects in cancer cells [47]. In addition to their single use, lactobacilli can be used in combination with other bacterial species, such as bifidobacterium, resulting in a broader clinical use. In summary, lactobacilli can be used in various medical fields; species or strain identification is crucial because different species or strains may lead to different therapeutic effects.

Despite the benefits of probiotics, their safety must be considered. Possible side effects or adverse events resulting from the use of probiotics include gastrointestinal symptoms such as abdominal cramping or diarrhea, skin manifestations such as rash or acne, excessive immune stimulation, and systemic infections such as bacteremia, endocarditis, and sepsis [48,49]. Systemic infections are generally considered the most severe, and patients who are immunosuppressed, critically ill, or have cancer are especially at high risk for these side effects [49]. In cases of bacteremia, Lactobacillus spp. strains were the most reported bacteria pathogen, including lactobacillus acidophilus, lactobacillus casei, and lactobacillus rhamnosus [48]. Several studies have also reported cases of lactobacillus spp.-induced bacteremia in patients with cancer [50,51]. However, in a systematic review that examined the safety of probiotics in cancer patients, no deaths could be attributed to their use [18]. In a phase II single-arm study published in 2016, the use of lactobacillus brevis CD2 reduced the incidence of oral mucositis in patients with hematological disorders undergoing high-dose chemotherapy plus HSCT; despite the immunosuppressed status of these patients, no positive blood cultures for lactobacillus brevis CD2 were identified [52]. In our included trials consisting of more than 350 cancer patients who received probiotics, no severe adverse events, such as bacteremia or even death, directly resulted from their use. In conclusion, the use of probiotics appears to be generally safe with a low incidence of adverse events over time. However, in patients with certain clinical conditions, such as those with immunosuppressed status, the possible risk following the use of probiotics should be carefully considered to balance benefit and harm.

Our study provided strong evidence on the effectiveness of probiotics against cancer therapy-induced oral mucositis. Probiotics significantly reduced the risk in the development of oral mucositis by 16% (RR = 0.84), and by 35% in that of severe oral mucositis (RR = 0.65) during cancer treatment. However, the protective effect of probiotics is inferior to other treatment options suggested by MASCC for cancer therapy-induced oral mucositis, such as honey, glutamine and photobiomodulation therapy, which can reduce the incidence of severe oral mucositis by up to 50–70% [1]. Considering the potential dose-response effect of probiotics on human bodies [53], their lower effectiveness for the development of oral mucositis might be attributed to relatively insufficient bacterial counts or prescribed frequency. In our opinion, probiotics still constitute an effective option for cancer therapy-induced oral mucositis; future trials in this field are still warranted to clarify the appropriate doses of each strain or species of bacteria for use as probiotics to achieve the best preventive and therapeutic effect on oral mucositis after cancer treatment.

In our study, the heterogeneity of the trials analyzed was low, suggesting low study variability and providing reliable evidence of the benefits of probiotics against oral mucositis. However, it should be known that the patients included in our trials were diagnosed with various types and different stages of cancers and treated with different doses or cancer therapy regimens, and the bacterial species or strains used as probiotics and the method of use were heterogeneous in the included trials.

This study has some limitations that should be addressed. First, the sample size included in this study was small. Second, because of the lack of complete data on body weight loss and QoL scores in our included trials, data pooling was not feasible. Third, our study only examined the effectiveness of lactobacillus spp. in preventing and mitigating oral mucositis; however, the possible effects of other bacterial species or strains are unknown. Fourth, as mentioned above, no current studies have focused on the doses, durations, and long-term effects of probiotic use in cancer patients, which warrants more trials to address this issue. Finally, no objective evidence indicates a low inflammatory response in the oral mucosa after the use of probiotics in cancer patients, and additional studies in this field should be conducted.

## 4. Materials and Methods

### 4.1. Selection Criteria

We identified and reviewed RCTs that investigated the effectiveness of probiotics in preventing and mitigating cancer therapy-induced oral mucositis. Studies that met the following criteria were included: (1) having a randomized controlled study design; (2) having full text available; (3) including patients with cancer receiving chemotherapy, radiotherapy, or any therapy that can lead to oral mucositis; (4) comparing the effects of probiotic use with those of either control or placebo; (5) providing a detailed description of the probiotics used and the method of usage; and (6) describing the definition and evaluation of oral mucositis severity. We excluded trials that met at least one of the following criteria: (1) having a nonrandomized controlled or single-arm study design, (2) including duplicate reporting of patient cohorts, and (3) not reporting clear outcomes or results.

### 4.2. Search strategy and Study Selection

This study was carried out in accordance with the Preferred Reporting Items for Systematic Reviews and Meta-analysis Guidelines [54]. The trials were identified by searching for keywords in PubMed, Embase, Cochrane Library, and ClinicalTrial.gov, and no language restrictions were applied. The following terms and the Boolean operator were used in MeSH and free-text searches: oral mucositis, cancer OR carcinoma OR malignancy OR tumor OR chemotherapy OR radiotherapy, probiotics OR lactobacillus OR bifidobacterium. The “related articles” option in PubMed was used to broaden the search. The last search was performed in September 2022. Additionally, we identified additional trials by manually searching the reference sections of relevant trials and contacting known experts in the related field. The PROSPERO registration number is 42022302339.

### 4.3. Data Extraction

Two reviewers (YCL and CRW) independently extracted the details of the RCTs related to the participants, inclusion and exclusion criteria, the probiotics used, and the incidence of oral mucositis. The individually recorded decisions of the two reviewers were compared, and disagreements were resolved by a third reviewer (TWH).

### 4.4. Methodological Quality Appraisal

The two reviewers independently appraised the methodological quality of each study based on the following criteria: (1) adequacy of randomization, (2) deviations from the intended intervention, (3) missing data on dropouts, (4) measurement of outcomes, and (5) selection of reported results [55].

### 4.5. Outcome Assessments

The primary outcomes of our study were (1) the incidence of oral mucositis and (2) the incidence of severe oral mucositis. The severity of oral mucositis was evaluated during cancer treatment. Subgroup analysis was performed to determine the effectiveness of certain bacterial species or strains used as probiotics and to evaluate patients’ responses to probiotics considering a certain type of cancer therapy or cancer, if necessary. Secondary outcomes were as follows: (1) enteral nutrition requirement during cancer treatment, (2) reduction in body weight, and (3) reported quality of life (QoL) of the patients.

### 4.6. Statistical Analysis

Meta-analysis was performed using Review Manager, version 5.4 (Cochrane Collaboration, Oxford, UK). The effect sizes of dichotomous outcomes are reported as risk ratios (RRs), with the precision of an effect size reported as a 95% confidence interval (CI). For a conservative statistical claim, the DerSimonian and Laird random-effect model was adopted for the calculation of pooled RRs. Statistical heterogeneity was examined using the I^2^ test, with I^2^ quantifying the proportion of total outcome variability attributable to variability between studies. Data were pooled only for trials with adequate clinical and methodological similarity; for trials whose data could not be pooled, qualitative analysis was performed.

## 5. Conclusions

Our study demonstrated the effectiveness of probiotics in the prevention and mitigation of cancer therapy–induced oral mucositis, and the use of probiotics generally appears to be safe. We recommend the use of probiotics to prevent and treat oral mucositis from the beginning of cancer treatment. Future studies investigating adequate doses of usage and the possible effects of different bacterial species or strains in cancer patients are warranted.

## Figures and Tables

**Figure 1 ijms-23-13268-f001:**
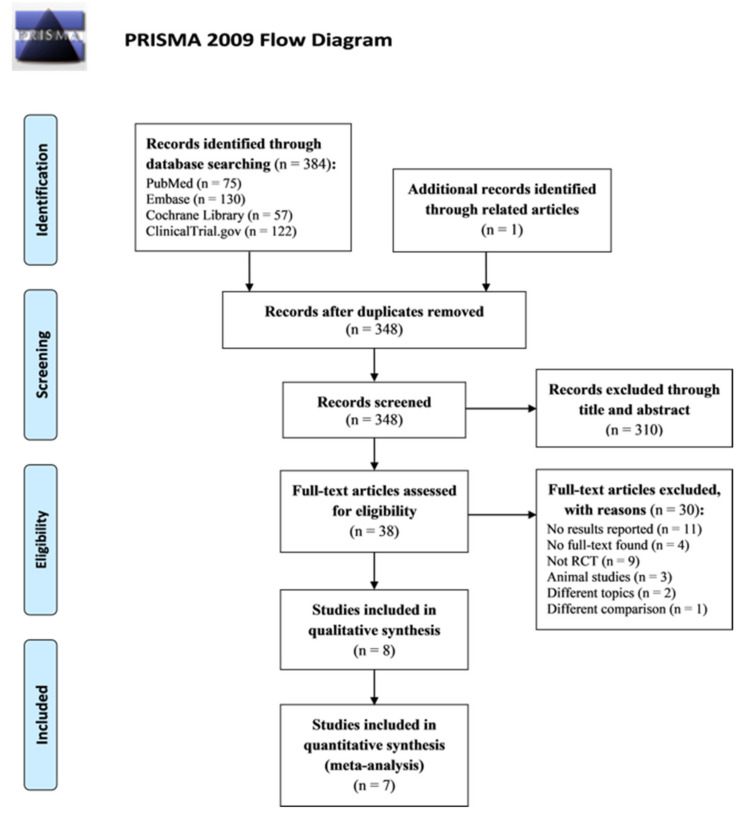
Flowchart of the selection of clinical trials.

**Figure 2 ijms-23-13268-f002:**
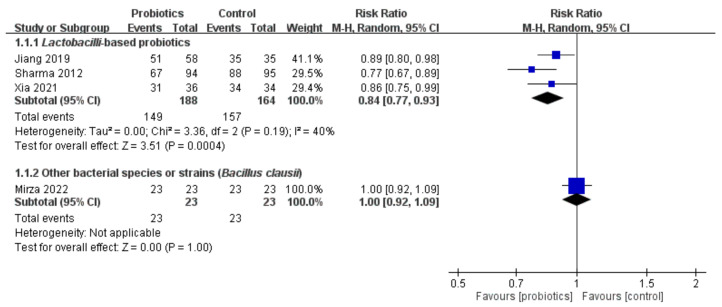
Forest plot of comparison: probiotics or control group. Outcome: incidence of oral mucositis. [17,23,25,27]

**Figure 3 ijms-23-13268-f003:**
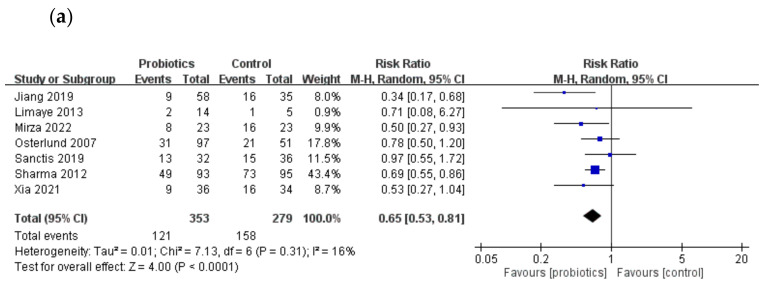
Forest plot of comparison: probiotics or control group. Outcome: (**a**) incidence of severe oral mucositis; (**b**) incidence of severe oral mucositis in patients receiving chemo-radiotherapy; (**c**) incidence of severe oral mucositis in patients receiving probiotics containing lactobacillus spp. only. [17,21,22,23,24,25,27]

**Table 1 ijms-23-13268-t001:** Characteristics of studies that fulfilled the inclusion criteria for meta-analysis.

Author [Year]	Inclusion Criteria	No. of Patients	Age, Year, Mean ± SD	Male (*n*, %)	Intervention
CT + RT					
Jiang,2019 [17]	Locally advanced NPC, without distant metastasis; aged 18–70 y; KPS ≥ 80; RT (70 Gy) and CT (cisplatin-based)	C: 35 I: 64	C: 50.40 ± 10.25 I: 51.69 ± 9.79	C: 21, 60% I: 37, 63.8%	C: placebo (starch), 3 capsules × 2 times/day × 7 weeks I: probiotic combination (*B. longum*, *L. lactis*, and *E. faecium*), 3 capsules × 2 times/day × 7 weeks
Mirza, 2022 [23]	HNC; aged between 30–60 y; RT (60–70 Gy) with or without CT (cisplatin-based)	C: 23 I: 23	55 (35–60) †	C: 21, 91.3% I: 22, 95.7%	C: placebo (distilled water 5 mL) × 2 times/day, until the completion of RT course I: *B. clausii* oral suspension, 5 mL × 2 times/day, until the completion of RT course
Sanctis, 2019 [21]	HNC (except larynx, parotid and other salivary glands tumors); aged ≥ 18 y; KPS > 70; RT (68–70 Gy) and CT (cisplatin-based; neo-adjuvant allowed for NPC)	C: 32 I: 36	C: 60 (39–77) † I: 58.4 (34–74) †	C: 27, 75% I: 26, 81.2%	C: sodium bicarbonate, at least 3 times/day, until the end of cancer treatment. I: *L. brevis* CD2 lozenges, 1 lozenge/2–3 h × 6 times/day to be dissolved in the mouth and then swallowed, up to 1 week after the end of the cancer treatment
Sharma, 2012 [25]	HNSCC stage II–IV; RT (70 Gy) and CT (cisplatin-based, 40 mg/m^2^)	C: 99 I: 101	C: 50.09 ± 10.04 I: 52.35 ± 9.43	C: 91, 91.9% I: 94, 93.1%	C: placebo (mixture of the sugars and salts), 1 lozenge/2–3 h × 6 times/day to be dissolved in the mouth and then swallowed × 8 weeks I: *L. brevis* CD2 lozenges, 1 lozenge/2–3 h × 6 times/day to be dissolved in the mouth and then swallowed × 8 weeks
Xia, 2021 [27]	Locally advanced NPC, without distant metastasis; aged 18–70 y; RT (70 Gy) and CT (cisplatin-based)	C: 38 I: 39	C: 51.70 ± 11.21 I: 52.61 ± 10.56	C: 11, 32% I: 11, 31%	C: placebo (no information), 1 capsule 2 times/day × 7 weeks I: probiotic combination (*L. plantarum* MH-301, *B. animalis* subsp. Lactis LPL-RH, *L. rhamnosus* LGG-18, and *L. acidophilus*), 1 capsule × 2 times/day × 7 weeks
**CT only**					
Limaye, 2013 [22]	Newly diagnosed HNSCC; scheduled to receive ≥ 2 cycles of induction CT (cisplatin-based)	C: 8 I1: 5 I2: 6 I3: 6	C: 54 (18–63) † I1: 61 (42–66) † I2: 54 (26–64) † I3: 52 (42–56) †	C: 7, 88% I1: 5, 100% I2: 2, 33% I3: 5, 83%	C: placebo rinse 15 mL × 6 times/day I1: AG013 oral rinse (containing *L. lactis*) 15 mL × 1 time/day I2: AG013 oral rinse (containing *L. lactis*) 15 mL × 3 times/day I3: AG013 oral rinse (containing *L. lactis*) 15 mL × 6 times/day
Österlund, 2007 [24]	Colorectal cancer, stage II-IV, s/p surgical resection; aged 18–75 y; ECOG ≤ 2; CT (leucovorin and 5-FU-based)	C: 52 I: 98	C: 57 (31–75) † I: 61 (35–74) †	C: 25, 48.1% I: 51, 52%	C: no information I: *L. rhamnosus* GG, 1 capsule × 2 times/day during the whole CT course
Topuz, 2008 [26]	Newly diagnosed colorectal cancer, stages II-IV; ECOG ≤ 2; CT (5-FU based, median: 6 cycle)	C: 20 I: 17	C: 58 (34–72) † I: 51 (19–75) †	C: 12, 60% I: 12, 70.6%	C: sodium chloride × 2 times/day, first 5 days of each CT cycle I: oral lavage with kefir (containing Lactobacillus spp., *Bifidobacterium* spp., etc.) and swallow 250 mL × 2 times/day after meal, first 5 days of each CT cycle

*B. clausii*, *Bacillus clausii*; *B. animalis*, *Bifidobacterium animalis*; *B. longum*, *Bifidobacterium longum*; CT, chemotherapy; DCF, docetaxel, cisplatin and 5-fluorouracil; ECOG, Eastern Cooperative Oncology Group performance status; *E. faecium*, Enterococcus faecium; FU, fluorouracil; Gy, Gray; HNC, head and neck cancer; HNSCC, head and neck squamous cell carcinoma; HSCT, hematopoietic stem cell transplantation; KPS, Karnofsky performance status; *L. acidophilus*, *Llactobacillus acidophilus*; *L. brevis*, *Actobacillus brevis*; *L. lactis*, *Lactobacillus lactis*; *L. rhamnosus*, *Lactobacillus rhamnosus*; NPC, nasal pharyngeal carcinoma; RT, radiotherapy; s/p, status post; y, years old †: median (range).

**Table 2 ijms-23-13268-t002:** Assessment of Methodological Quality of Included Trials (RCT evaluated by ROB 2.0).

Author [Year]	Bias Caused by Adequacy of Randomization	Bias Caused by Deviations from Intended Interventions	Bias Caused by Missing Data of Dropouts	Bias in Measurement of the Outcomes	Bias in Selection of the Reported Results	Overall Risk of Bias
Jiang 2019 [17]	Low risk	Low risk	Low risk	Low risk	Low risk	Low risk
Limaye 2013 [22]	Some concerns ^1^	Low risk	High risk ^2^	Some concerns ^3^	Low risk	High risk
Mirza 2022 [23]	Low risk	Low risk	Low risk	Low risk	Low risk	Low risk
Österlund 2007 [24]	High risk ^4^	Low risk	Low risk	Low risk	Low risk	High risk
Sanctis 2019 [21]	High risk ^5^	High risk ^6^	Low risk	Low risk	Low risk	High risk
Sharma 2012 [25]	Low risk	Low risk	Low risk	Low risk	Low risk	Low risk
Topuz 2008 [26]	Some concerns ^7^	Low risk	Low risk	Low risk	Low risk	Some concerns
Xia 2021 [27]	Low risk	Low risk	Low risk	Low risk	Low risk	Low risk

^1^: no information and detail on the method of randomization; ^2^: high percentage of missing data (24%); ^3^: no information on the blindness of the outcome assessor; ^4^: open-label design; ^5^: open-label design; ^6^: high percentage of interruption of probiotic intake (34.3%); ^7^: no information and detail for the way of randomization.

## Data Availability

All data generated or analyzed during this study are included in this published article.

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
