# Peer review of "Preventive Effect of Probiotics on Oral Mucositis Induced by Cancer Treatment: A Systematic Review and Meta-Analysis"

_ijms, 2022, doi:10.3390/ijms232113268_

Round 1
Reviewer 1 Report
The study is a systematic review and meta-analysis study of randomized controlled trials that aim to evaluate the preventive and therapeutic effect of probiotics in the development of oral mucositis induced by cancer therapy. The study is interesting and well-designed, and the manuscript is well-written. There are some minor issues that need to be considered.
- On the title, it is recommended to add “systematic review”
- The authors referred to the presence of a similar recent systematic review and the aim of their study is to make an update to it. It is not clear the kind of update has been performed with their study. It is recommended to declare the kind of update that would be addressed with their study in particular because the previous similar study is quite recent.
- The position of figures 1, 2, and 3 and tables 1, and 2 are far from their citation in the text. It is better to reposition them just after being cited in the text to avoid confusion for the readers.
- Figure 1 is very small in size.
- On table 2, it appears that the table footer is not provided, Please revise it.
- In the discussion section, as the authors referred to the presence of a previous similar study, it is recommended to discuss and compare their results with the results of this previous study and other older studies.
- In the discussion section, it would be better to compare the results of this study with the results of other systematic studies that evaluate other modalities for the prevention and therapy of oral mucositis such as photo-biomodulation (PBM).
- In the methods section, the authors declared that there were no language restrictions for the inclusion of the studies in this systematic review. It is preferred to declare how many studies were included in other languages than English and what are the methods for the translation.
Reviewer 2 Report
Patients and caregivers are highly motivated to reduce side effects of chemotherapy and radiation. Use of probiotics is a common question in the oncology clinic. This paper provides a high level of support for use of lactobacillus probiotics as one means to ameliorate mucositis. Two additional potential references to cite in the discussion for other means to reduce mucositis and to promote better eating are attached. Publication should be of interest to dieticians, pediatric and medical oncologists.
